Metabolomics of a cell line-derived xenograft model reveals circulating metabolic signatures for malignant mesothelioma

Gao Yun 1 2 3
Dai Ziyi 1 2
Yang Chenxi 1 2
Wang Ding 1 2 4
Guo Zhenying 1 2
Mao Weimin maowm@zjcc.org.cn 1 2 3
Chen Zhongjian chenzj@zjcc.org.cn 1 2 3
1 The Cancer Hospital of the University of Chinese Academy of Sciences (Zhejiang Cancer Hospital) , Hangzhou , China
2 Institute of Basic Medicine and Cancer (IBMC), Chinese Academy of Sciences , Hangzhou , China
3 Zhejiang Key Laboratory of Diagnosis&Treatment Technology on Thoracic Oncology(Lung and Esophagus) , Hangzhou , China
4 Zhejiang Chinese Medical University , Hangzhou , China
Felley-Bosco Emanuela
Electronic publication date: 2022 Jan 4
Publication date: 2022
Volume: 10
Electronic Location ID: e12568
Received 2021 May 21; Accepted 2021 Nov 8
Copyright: ©2022 Gao et al.
Copyright year: 2022
Copyright holder: Gao et al.
License: This is an open access article distributed under the terms of the Creative Commons Attribution License, which permits unrestricted use, distribution, reproduction and adaptation in any medium and for any purpose provided that it is properly attributed. For attribution, the original author(s), title, publication source (PeerJ) and either DOI or URL of the article must be cited.
License URL: https://creativecommons.org/licenses/by/4.0/

Keywords: Cell line-derived xenograft, GC-MS, Malignant mesothelioma, Metabolomics, Amino acid metabolism

Funding: The National Natural Science Foundation of China 81672315 81302840 82072577 Key R&D Program Projects in Zhejiang Province 2018C04009 Natural Science Foundation of Zhejiang Province of China LY21H160002 LY20H280001 Excellent Youth Talent Program of Traditional Chinese medicine of Zhejiang province science and technology plan project 2017ZQ007 This work was supported by the National Natural Science Foundation of China (Grant numbers: 81672315 & 81302840 & 82072577), Key R&D Program Projects in Zhejiang Province (Grant number 2018C04009), Natural Science Foundation of Zhejiang Province of China (Grant numbers: LY21H160002 & LY20H280001), Excellent Youth Talent Program of Traditional Chinese medicine of Zhejiang province science and technology plan project (Grant number: 2017ZQ007). The funders had no role in study design, data collection and analysis, decision to publish, or preparation of the manuscript.

==============================
Background

Malignant mesothelioma (MM) is a rare and highly aggressive cancer. Despite advances in multidisciplinary treatments for cancer, the prognosis for MM remains poor with no effective diagnostic biomarkers currently available. The aim of this study was to identify plasma metabolic biomarkers for better MM diagnosis and prognosis by use of a MM cell line-derived xenograft (CDX) model.

Methods

The MM CDX model was confirmed by hematoxylin and eosin staining and immunohistochemistry. Twenty female nude mice were randomly divided into two groups, 10 for the MM CDX model and 10 controls. Plasma samples were collected two weeks after tumor cell implantation. Gas chromatography-mass spectrometry analysis was conducted. Both univariate and multivariate statistics were used to select potential metabolic biomarkers. Hierarchical clustering analysis, metabolic pathway analysis, and receiver operating characteristic (ROC) analysis were performed. Additionally, bioinformatics analysis was used to investigate differential genes between tumor and normal tissues, and survival-associated genes.

Results

The MM CDX model was successfully established. With VIP > 1.0 and P-value < 0.05, a total of 23 differential metabolites were annotated, in which isoleucine, 5-dihydrocortisol, and indole-3-acetamide had the highest diagnostic values based on ROC analysis. These were mainly enriched in pathways for starch and sucrose metabolism, pentose and glucuronate interconversions, galactose metabolism, steroid hormone biosynthesis, as well as phenylalanine, tyrosine and tryptophan biosynthesis. Further, down-regulation was observed for amino acids, especially isoleucine, which is consistent with up-regulation of amino acid transporter genes SLC7A5 and SLC1A3 in MM. Overall survival was also negatively associated with SLC1A5, SLC7A5, and SLC1A3.

Conclusion

We found several altered plasma metabolites in the MM CDX model. The importance of specific metabolic pathways, for example amino acid metabolism, is herein highlighted, although further investigation is warranted.

Introduction

Malignant mesothelioma (MM) is an uncommon but highly aggressive tumor that is associated with asbestos exposure. The worldwide, age-adjusted mesothelioma mortality rate has increased approximately 5.37% annually (Delgermaa et al., 2011). It is expected that the age-adjusted mesothelioma incidence and related mortality rate will continue to increase dramatically in the near future, especially in countries where asbestos is still widely used (e.g., China, India, and the Russian Federation) (Carbone et al., 2019). The median survival time for patients with MM is less than 12 months from onset (Ledda, Senia & Rapisarda, 2018). Treatment is challenging for several reasons, but most important is the difficulty of early stage diagnosis when the patient is typically asymptomatic. In addition, symptoms tend to be vague and often resemble those of other more common diseases such as chest infection (Ahmadzada, Reid & Kao, 2018). As a result, MM diagnosis is often delayed resulting in inevitable tumor development. Therefore, MM patients urgently need accurate, susceptive, and non-invasive procedures by which to predict, diagnose, and prognosticate disease outcomes.

Although cancer is traditionally viewed as a disease of cellular proliferation, more recent studies have proposed cancer as a metabolic disease (Seyfried et al., 2014). Tumors, being highly proliferative, show significant alteration in metabolic pathways, especially in energy production and the biosynthesis of macromolecules (Hammoudi et al., 2011). Therefore, metabolites such as peptides, fatty acids, and steroids in tissues and body fluids (e.g., urine and serum) can provide insight into important disease characteristics.

Metabolomics provides a comprehensive metabolite profile of any biological sample, detecting metabolites involved in the pathophysiology of a disease and serving as a guide for the identification of useful biomarkers (Clish, 2015). The most popular methods for metabolomics analysis are gas chromatography-mass (GC-MS), liquid chromatography-mass spectrometry (LC-MS), and nuclear magnetic resonance (NMR) spectroscopy. Metabolomics has the capacity to detect thousands of feature ions at once, which has made it increasingly popular in the cancer research field for biomarker identification (Beger, 2013). In one study of Asian triple negative breast cancer patients, global metabolomics identified altered metabolites that enabled the construction of metabolite-based biomarker panels (Li et al., 2020). As such, metabolomics can be a powerful and useful tool for cancer research.

However, there are few metabolomics studies of MM. Therefore, the aim of this study was to discover promising metabolite biomarkers to improve early MM diagnosis. Such biomarkers would allow for customized and individualized patient treatment increasing overall survival. In addition, potential survival-related genes were explored by bioinformatics analysis based on the TCGA database. Identified biomarkers and genes may prove to be promising therapeutic targets for the treatment of MM.

Materials and Methods

MM cell line

Ren cells, a human MM cell line established by Smythe et al. (1994), were cultured in Gibco™ Dulbecco’s modified eagle medium (DMEM) with L-glutamine (Thermo Fisher Scientific, Waltham, MA, USA), supplemented with 10% Fetal Bovine Serum (FBS) (Thermo Fisher Scientific), 100 units/ml penicillin, and 100 µg/mL streptomycin (Thermo Fisher Scientific). Cells were incubated at 37 °C in a 5% CO2 humidified atmosphere. Trypsin was used for cell collection when confluence reached 80%.

Xenograft model construction

Animal experimentation was performed under project license (No. 2019-02-010) granted by the Institutional Animal Care and Ethics Committee of Zhejiang Cancer Hospital, in compliance with national or institutional guidelines for the care and use of animals. All animals were housed in individual cages (five mice per cage), with full-value nutritional granulated fodder and distilled water, at 21 ± 2 °C, at 40%–60% relative humidity, with 12-h light and dark periods.

Twenty female BALB/c nude mice (4-week-old) were purchased from Shanghai SLAC laboratory Animal Co., Ltd. (Shanghai, China) and acclimated for one week before experimentation. The mice were randomly divided into two groups of 10. For the cell-derived xenograft (CDX) model, 200 µL of PBS containing 3 × 106 cells was subcutaneously injected into the flanks of mice. The control group received no treatment. Two weeks after transplantation, blood was obtained from the retro-orbital plexus with isoflurane-induced anesthesia. A blood volume of 100 µL was collected into a heparinized tube, plasma was separated at 3,000 rpm for 15 min at 4 °C, and samples were stored at −80 °C until analysis. The anesthesia protocol was that provided with the anesthesia machine produced by RWD Life Science Co., Ltd. (GuangDong, China). After experimentation, animal carcasses were loaded into garbage bags and handled by the Institute of Laboratory Animals.

Immunohistochemistry (IHC)

Tumors were collected and preserved in 4% paraformaldehyde. IHC was performed with formalin-fixed paraffin-embedded (FFPE) tissue of tumor samples. Slides were deparaffinized and incubated in 3% hydrogen peroxide to inactivate endogenous peroxidases. Slides were placed into citric acid repair solution (pH = 6) and boiled at 100 °C for 90 s for antigen retrieval. After addition of a protein blocking solution, slides were incubated with antibodies (calretinin, CK5/6, WT1, D2-40, MOC31, SLC1A5, SLC7A5) at 4 °C overnight, and then incubated with HRP-labeled secondary antibody (Agilent DAKO, CA, USA, catalogue number #K5007), then the slides were further processed with the DAB regent kit (Agilent DAKO, CA, USA, catalogue number #K5007). The following antibodies were used: mouse anti calretinin monoclonal antibody (Leica Biosystems, Wetzlar, Germany, catalogue number #PA0346), mouse anti CK5/6 monoclonal antibody (MXB Biotechnologies, Fujian, China, catalogue number # MAB-0744), mouse anti WT1 monoclonal antibody (catalogue number #IR055; Agilent DAKO, Santa Clara, CA, USA), mouse anti podoplanin (D2-40) monoclonal antibody (catalogue number #IR072; Agilent DAKO), mouse anti MOC31 monoclonal antibody (MXB Biotechnologies, Fujian, China, catalogue number # MAB-0280), rabbit anti SLC1A5 monoclonal antibody (catalogue number #8057; (Cell Signaling Technology, Danvers, MA, USA), and rabbit anti SLC7A5 polyclonal antibody (catalogue number # 13752-1-AP; Proteintech, Rosemont, IL, USA). Histological features of MM were identified based on the guidelines for the diagnosis and treatment of pleural MM (Van Zandwijk et al., 2013). Immunohistochemical analysis was performed on FFPE tissue specimens from MM patients after ethical committee approval (IRB-2018-82).

Gas Chromatography-Mass Spectrometry (GC-MS)-based metabolomics profiling

Sample preparation

An aliquot of 50 µL plasma was mixed with 200 µL of methanol (HPLC grade; CNW Technologies, Duesseldorf, Germany) containing 25 µmol/L 2-chloro-L-phenylalanine (Shanghai Hengbai Biotech, Shanghai, China) as an internal standard, which was used for normalization. After vortexing for 30 s, samples were ultra-sonicated in iced water for 10 min. After centrifugation at 12 000 rpm at 4 °C for 15 min, 180 µL of supernatant for each sample were transferred into a new tube and dried in a vacuum concentrator. The residues of each sample were then added with 20 µL of 20 mg/mL methoxyamine hydrochloride (AR grade; TCI, OR, USA) in pyridine (HPLC grade; Adamas Pharmaceuticals, CA, USA). The above samples were kept at 80 °C for 30 min, after which 40 µL of bis-(trimethylsilyl) trifluoroacetamide (BSTFA) in 1% TMCS (REGIS Technologies, IL, USA) was added and samples were incubated at 70 °C for 1.5 h. The samples were stored at room temperature until analysis. Quality control (QC) samples were prepared by pooling an aliquot of 30 µL from each sample.

GC-TOF-MS analysis

Metabolomics was performed on an Agilent 7890 gas chromatograph system equipped with Pegasus HT time-of-flight mass spectrometer (LECO, St. Joseph, MI, USA). Chromatographic separation was achieved on a DB-5MS capillary column (30 m ×0.25 mm, 0.25 µm film thickness; J&W Scientific, Folsom, CA, USA). The GC-TOF-MS settings were as follows: 1 µL sample was loaded in a splitless mode, 3 mL/min for the front inlet purge flow rate, 1 mL/min for the gas flow rate; the initial temperature was 50 °C, then increased to 310 °C with a rate of 20 °C/min, then kept at 310 °C for 6 min; the temperatures for injection, transfer line, and ion source were 280, 280, and 250 °C, respectively; the electron impact energy was set at 70 eV; full-scan mode with a mass-to-charge ratio range from 50 to 500 was used in mass collection.

Metabolomics data analysis

GC-TOF-MS raw data were processed by Chroma TOF 4.3X software (LECO Corporation, St. Joseph, MI, United States). The LECO-Fiehn Rtx5 database was used for data extraction and analysis. Normalization was performed for each sample through calculating the ratios of peaks of analytes and 2-chloro-L-phenylalanine, and relative quantification was applied in further comparison between xenograft and control groups. Metabolite identification was achieved by matching both mass spectra and retention index of the detected metabolic features. Peaks that were not detected in more than half of QC samples or peaks with RSD > 30% in QC samples were removed, a dataset with 20 samples and identified metabolites was obtained.

Both univariate and multivariate analysis were conducted to select the differential metabolites. Log2 (Fold change) (Log2(FC)) of each metabolite between xenograft and control groups were calculated through dividing their average values. -Log10 (P-value) was calculated for each metabolite using a two-tailed Student’s t-test. Besides, principal component analysis (PCA) of all the identified metabolites was used to observe the clusters of samples. Partial least squares-discriminant analysis (PLS-DA) was performed to select the most significant differential metabolites between xenograft and controls using R package “ropls” (version 1.18.8), and variable importance in projection (VIP) of each metabolite was obtained from PLS-DA. Permutation test with cross validation for 20 times was applied to test reliability of the PLS-DA model. Volcano plot with Log2(FC) and -Log10 (P-value) values was used to show the differential metabolites between xenograft and control groups.

Hierarchical clustering analysis and metabolic pathway analysis

Hierarchical clustering analysis was performed based on relative abundance of differential metabolites between MM and control groups. Results were illustrated as a heatmap using R package “pheatmap”. Pathway analysis of the differential metabolites was performed with an online tool (https://www.metaboanalyst.ca/). The top 25 pathways with P-value <0.05 were selected based on enrichment ratio (i.e., observed hits/expected hits).

Correlation analysis and receiver operating characteristics (ROC) analysis

Pearson correlation analysis was performed for each pair of differential metabolites, the results were visualized using R package “corrplot”.

ROC analyses were performed for all selected differential metabolites using all samples. Based on area under curve (AUC), top 6 metabolites were selected and their relative levels in MM and control were shown as box plots. R package “pROC” was used for ROC analyses and package “ggplot2” was used for visualization.

Bioinformatics analysis for amino acid transporter genes

The gene array dataset (GSE51024) from 55 MM tumor samples and along with paired normal tissue (for 41 tumors) was obtained from the GEO database (https://www.ncbi.nlm.nih.gov/geo). Statistical analysis was performed using R package “limma” to screen for statistically differentially expressed genes (P-value <0.05) and to calculate fold changes of selected genes. mRNA expression in type of RNA-seq count and corresponding survival data for 86 MM patients from Cancer Genome Atlas (TCGA) were downloaded from GDC TCGA mesothelioma dataset in the UCSC Xena TCGA hub (https://xenabrowser.net/). Amino acid transporter gene list was based on Bröer (2020). Samples were divided into high and low groups according to median value of each gene as cut-off value. Kaplan–Meier plots with log-Rank test were performed to determine significant differences between the two groups. Finally, univariate Cox proportional hazards regression analysis was performed to determine relationships among amino acid transporter genes and prognosis. P-value <0.05 was considered significant.

Results

CDX model information

Hematoxylin and eosin (H&E) staining showed REN cell line-derived xenografts to be biphasic MM, characterized by both epithelioid cell structure and sarcomatous components (Fig. 1A). IHC characteristics were consistent with MM clinico-pathological features, including diffuse positivity for calretinin (Fig. 1B) and focal positivity for WT1 (Fig. 1C) for mesothelioma, focal positivity for CK5/6 (Fig. 1D), wide positivity for D2-40 (Fig. 1E) for epithelial carcinoma, and negative for MOC31 (Fig. 1F) adenocarcinoma. The results demonstrate the MM cell line-derived xenograft model to be successfully established.

Metabolic differences between the MM xenograft group and the control group

A total of 377 peaks were obtained, and 187 metabolites were annotated for further multivariate analysis. The score plots of the first two principal components (PC1/PC2) in PCA showed an overlap between the xenograft model and control group, while PLS-DA analysis revealed a clear separation between the xenograft model and control, with a R2Y value of 0.910 and a Q2 value of 0.371 (Figs. 2A, 2B). Permutation tests further revealed the current PLS-DA model to be reliable (Fig. 2C). After application of filtering criteria, a total of six up-regulated metabolic features and 42 down-regulated metabolic features were obtained (Fig. 2D).

Figure 1 Histopathology (HE) and immunohistochemistry (IHC) findings in tumors from malignant mesothelioma xenograft mice.

(A) Representative images of H&E staining, the tumor cells were morphologically diverse, large in size, and markedly heteromorphic. (B–C) IHC staining with mesothelioma markers Calretinin and WT1 were diffuse positive and focal positive. (D–E) IHC staining with epithelial carcinoma markers CK5/6 and D2-40 were focal positive and widely positive. (F) IHC staining of adenocarcinoma marker MOC31 was negative. (magnification: HE, ×400; IHC, ×400).

Figure 2 Multivariate statistical analysis of metabolites.

(A) Score scatter plot of PCA analysis for plasma metabolomics between xenograft mice and controls. T represent the xenograft mice group, and N is the control group. (B) Score scatter plot of PLS-DA model for group xenograft mice and controls. The abscissa PC1 represents the first principal component, and the ordinate PC2 represents the second principal component. (C) Permutation test of PLS-DA modeling. (D) Volcano plots showing differential feature ions in model mice vs. controls. Differential ions were defined by VIP > 1.0 and P-value < 0.05; ions with Log2(Fold Change) > 0 were considered as upregulated, vice versa. Significantly up-regulated ions are indicated in red, significantly down-regulated ions are indicated in blue, and non-significant differences in metabolites are grey.

Differential metabolic patterns and pathway enrichment

After annotation, 23 metabolites were obtained, including 6 up-regulated and 17 down-regulated (Table 1). Based on these differential metabolites, the CDX group showed a distinctive metabolic pattern that differed from the control group (Fig. 3A). Based on all differential metabolites, 19 metabolic pathways were enriched. Steroid hormone biosynthesis was the most significantly enriched pathway with three hits, followed by starch and sucrose metabolism, pentose and glucuronate interconversions, as well as phenylalanine, tyrosine, and tryptophan biosynthesis pathways (Table 2).

Table 1 List of identified differential metabolites between model group and healthy controls.

No	Metabolite	VIP a	P-value	Fold change b	AUC ROC c	
1	Isoleucine	1.9	4.70E−03	0.48	0.860	
2	5-Dihydrocortisol	2.1	8.10E−03	0.26	0.860	
3	Guanosine	1.8	8.80E−03	0.31	0.850	
4	Cholesterol	1.8	9.60E−03	0.75	0.800	
5	4-Hydroxyphenylacetic acid	1.8	1.13E−02	0.39	0.810	
6	Cortexolone	1.7	1.42E−02	0.53	0.800	
7	Diglycerol	1.8	1.49E−02	1.54	0.830	
8	Trehalose-6-phosphate	1.7	1.57E−02	0.45	0.840	
9	Indole-3-acetamide	1.7	1.72E−02	1.42	0.850	
10	Glucose-1-phosphate	1.6	1.75E−02	1.64	0.820	
11	Sophorose	1.6	2.04E−02	0.46	0.800	
12	2-Monoolein	1.7	2.16E−02	0.3	0.820	
13	Hydrocortisone	1.6	2.23E−02	0.45	0.780	
14	Leucrose	1.6	2.24E−02	0.46	0.770	
15	Citric acid	1.6	2.54E−02	0.68	0.750	
16	Trehalose	1.5	2.79E−02	0.27	0.775	
17	Naringenin	1.5	3.55E−02	0.45	0.785	
18	Tyrosine	1.5	3.55E−02	1.58	0.770	
19	Cyclohexane-1,2-diol	1.6	3.99E−02	47.19	0.655	
20	Xylitol	1.5	4.10E−02	0.68	0.820	
21	24,25-Dihydrolanosterol	1.5	4.13E−02	0.59	0.780	
22	Indolelactate	1.5	4.21E−02	0.54	0.730	
23	Pentadecanoic acid	1.4	4.47E−02	2.48	0.640	
Notes.

a Variable Importance in Projection (VIP) values from Partial least squares-discriminant analysis (PLS-DA).

b Fold change of metabolites expression in model group compared to controls.

c ROC, receiver-operating characteristic curve; AUC, area under the ROC curve.

Figure 3 Heatmap from hierarchical clustering analysis and a diagram of the metabolic pathway enrichment analysis.

(A) Heatmap comparing levels of the metabolites of the model mice group and the control group. A single mouse corresponded to each column of the heatmap. The mice of the model group are labeled by red color (T), and the controls are blue (N). The color scale indicates the relative expression levels of the metabolites across all samples, blue represents an expression less than the mean, while red represents a higher expression level greater than the mean. (B) Pathway analysis of the metabolites in the model mice group.

Table 2 Clustered metabolic pathways.

Metabolism pathway	Total	Hits	P-value	
Starch and sucrose metabolism	18	2	1.75E−02	
Pentose and glucuronate interconversions	18	2	1.75E−02	
Galactose metabolism	27	2	3.77E−02	
Phenylalanine, tyrosine and tryptophan biosynthesis	4	1	4.57E−02	
Steroid hormone biosynthesis	85	3	7.15E−02	
Steroid biosynthesis	42	2	8.34E−02	
Tyrosine metabolism	42	2	8.34E−02	
Valine, leucine and isoleucine biosynthesis	8	1	8.94E−02	
Ubiquinone and other terpenoid-quinone biosynthesis	9	1	1.00E−01	
Aminoacyl-tRNA biosynthesis	48	2	1.05E−01	
Phenylalanine metabolism	10	1	1.11E−01	
Glycerolipid metabolism	16	1	1.71E−01	
Citrate cycle (TCA cycle)	20	1	2.09E−01	
Glycolysis / Gluconeogenesis	26	1	2.64E−01	
Alanine, aspartate and glutamate metabolism	28	1	2.81E−01	
Glyoxylate and dicarboxylate metabolism	32	1	3.14E−01	
Amino sugar and nucleotide sugar metabolism	37	1	3.54E−01	
Valine, leucine and isoleucine degradation	40	1	3.77E−01	
Primary bile acid biosynthesis	46	1	4.20E−01	
Purine metabolism	65	1	5.40E−01	
Notes.

Metabolomics pathway data analysis was performed by MetaboAnalyst.

Diagnostic value of plasma metabolites

Figure 4A shows the Spearman correlations for the differential metabolites based on their relative circulating levels. ROC analysis identified the top six metabolites with the highest AUCROC values. They were isoleucine (AUC: 0.860), 5-dihydrocortisol (AUC: 0.860), guanosine (AUC: 0.850), indole-3-acetamide (AUC: 0.850), trehalose-6-phosphate (AUC: 0.840), and diglycerol (AUC: 0.830) (Figs. 4B–4G). Of these, indole-3-acetamide and diglycerol were up-regulated in the MM xenograft model, with the other four down-regulated.

Figure 4 Metabolite-based correlation analyses and ROC analyses.

(A) Correlation pattern among measured metabolites over the entire cohort using pearson correlation. (B–G) Receiver operating characteristic (ROC) curves of the six differential metabolites with highest area under curve (AUC) values: Isoleucine, lyxonic acid, 5-dihydrocortisol, guanosine, indole-3-acetamide, and trehalose-6-phosphate; and boxplots comparing relative intensities of these metabolites between cell-derived xenograft (CDX) (T) and control groups (N). FC: fold change; P-value was calculated from a two-tailed Student’s t-test, and P-value < 0.05 was considered as significant.

Dysregulated amino acid metabolism

In the plasma of CDX model mice, most of the detected amino acids were downregulated including isoleucine, valine, and proline, with the exception of tyrosine (Fig. 5A). Isoleucine, which was decreased in the mesothelioma model, had diagnostic value (Fig. 4B).

Figure 5 Differential amino acids in MM and survival-related transporters.

(A) Heatmap comparing the amino acids of samples in the model mice group (T) and the controls (N). The color scale indicates the relative expression levels of the amino acids, blue represents an expression level less than the mean, while red represents an expression level greater than the mean. (B–D) Kaplan–Meier survival curves for patients with mesothelioma grouped by the different expression levels of SLC1A5 (B), SLC7A5 (C), SLC1A3 (D).

The bioinformatics analysis identified many prognosis-related genes. These genes were compared to a list of amino acid transporter genes in the referenced paper, with overlap mainly with SLC transporter genes. Survival analysis of the SLC genes indicated that the three most significant genes were SLC1A5 (HR = 1.70), SLC7A5 (HR = 2.35), and SLC1A3 (HR = 2.21) (Fig. 5B). All of the most significant SLC genes had hazard ratios over one, indicating them to be unfavorable prognostic factors (Table 3). Data from GEO (Table S1) showed that gene expressions of amino acid transporters SLC7A5 and SLC1A3 were significantly upregulated, whereas SLC1A5 was downregulated in tissues of MM patients compared to controls. Immunohistochemistry also validated expressions of SLC1A5 and SLC7A5 in MM tumor specimens from both mouse xenografts (Figs. S1A), S1C) and MM patients (Figs. S1B, S1D).

Discussion

By using a MM cell line, we constructed a xenograft model with pathological characteristics highly consistent with that of MM. Strong positive expression of calretinin and negative expression of MOC31 are compatible with the pathological features of MM cases (Van Zandwijk et al., 2013). Thus, the Ren cell line xenograft model is a reliable tool, readily available for research purposes.

Oncogenic mutations in tumor cells can induce a metabolic imbalance in native homoeostasis. Our study, based on GC-MS metabolomics, identified 23 differentially expressed plasma metabolites for the CDX group and the control group. The PLS-DA model showed good separation for the model group and the healthy control group. The most notable metabolites are discussed below.

It is widely acknowledged that some cancers are hormone-dependent (e.g., breast and prostate cancer) (Key, 1995), whereas no study reported hormones as carcinogens for MM. Nevertheless, we detected changes in plasma levels of some hormones. We observed 5-dyhydrocortisol and hydrocortisone levels to be significantly decreased. Hydrocortisone is a steroid hormone that is typically anti-inflammatory and is used for treatment of cancers, such as leukemia. Asbestos can induce inflammatory changes, including increase in MMP7, CXCR5, CXCL13, and CD44, and this chronic inflammation can lead to chronic diseases, such as cancer (Kumagai-Takei et al., 2018). During chronic inflammation, pro-inflammatory metabolites are upregulated to create an ideal environment for tumour growth (Rayburn, Ezell & Zhang, 2009). Therefore, we speculate that hydrocortisone exhibits an anti-tumour effect by reducing the inflammatory response within the tumor (Linton et al., 2012).

It should be noted that cholesterol, which can be used to produce hydrocortisone, was also decreased. This supports our speculation that hydrocortisone was depleted in diseased tissue. Meta-analysis studies conducted on patients with low cholesterol have shown an association with low cholesterol levels and a high risk for cancer. However, a causal relationship has not been established. Low cholesterol may not be a risk factor for cancer, rather it may be a sign of undiagnosed cancer (Tanne, 2007). Therefore, we speculate that cancer can lower cholesterol levels. A decrease in cholesterol levels may be a result of increased uptake by tumour cells. Cholesterol is a key component of cell membranes and is essential to cell proliferation (Fernández, De Cedrón & De Molina, 2020). Another significantly altered lipid was glycerol. Glycerol is component of triacylglycerol. In colorectal cancer, total lipid content is reduced (Mika et al., 2020). Herein, we speculate that tumor cells dysregulate lipid metabolism, leading to increased oxidation of triacylglycerol in order to produce more energy for tumor growth, consequently releasing more glycerol into the circulation.

Isoleucine is also decreased significantly. Isoleucine is an essential amino acid (EAA) for mammals, used for biosynthesis, energy production, and as a mediator of redox balance (Lieu et al., 2020). Amino acids can provide both nitrogen and carbon for necessary biosynthesis of molecules such as purines as well as serve as an alternative to glucose for the energy necessary in tumor development and cancer cell proliferation. Amino acids are also involved in activities such as NADPH production and glutathione synthesis, essential to redox hemostasis in cancer cells (Vučetić et al., 2017). Isoleucine was also reported to upregulate in pancreatic ductal adenocarcinoma (PDAC) (Jiang et al., 2021), so we speculate that the decrease in isoleucine in our study is due to constant cancer cell uptake. The high demand for EAAs is also reflected in the upregulation of EAA transporters that are evident in many cancers (Lieu et al., 2020). In MM, survival analysis of amino acid transporter genes SLC1A5, SLC7A5, and SLC1A3 were found to be unfavorable prognostic factors, consistent with previous findings. In addition, expressions of SLC7A5 and SLC1A3 were significantly upregulated in MM based on bioinformatic analysis, indicating its prominence to be a therapeutic target. On contrary, SLC1A5 was downregulated, which is inconsistent with results from other studies, therefore more research is needed to validate this trend. Isoleucine is an important branched chain amino acid (BCAAs). Such amino acids are essential for protein synthesis, as a source of nitrogen, and for energy production, all of which are vital for tumor growth (Ananieva & Wilkinson, 2018). BCAAs can also donate nitrogen for glutamate synthesis, which can then be used to synthesize glutamine, important to tumor growth and maintenance (Hutson, Sweatt & LaNoue, 2005). The process is catalyzed by branched-chain aminotransferase 1 (BCAT1) and mitochondrial branched-chain aminotransferase 2 (BCAT2). Recent evidence has shown that these enzymes are overexpressed in many cancers (Ananieva & Wilkinson, 2018). Survival analysis of BCAT1 by GEPIA at a quartile cut-off level demonstrated a significant difference between the high and low expression groups. High expression was associated with a poor prognosis, consistent with our findings.

Table 3 Survival associated SLC transporters genes.

Gene	P-value a	HR b	HR_low c	HR_high d	
SLC1A5	2.54E−02	1.70	1.06	2.72	
SLC7A5	3.71E−04	2.35	1.45	3.82	
SLC1A3	7.97E−04	2.21	1.38	3.56	
SLC6A9	1.23E−02	0.55	0.34	0.88	
SLC43A2	2.01E−03	0.47	0.29	0.77	
SLC1A7	3.97E−03	0.49	0.30	0.80	
SLC7A8	8.41E−03	0.53	0.33	0.86	
SLC43A3	5.30E−04	2.29	1.42	3.70	
SLC38A3	1.46E−02	1.80	1.12	2.89	
Notes.

a Kaplan-Meier survival analysis of SLC transporters genes, significance was set at P < 0.05.

b HR, hazard ratio.

c Lowest confidence interval.

d Highest confidence interval. A confidence interval of 95% was used.

In this study, we found that citrate levels were significantly decreased in the cancer model group compared to the control group. Recent studies have suggested elevated consumption of citrate is a consequence of higher metabolic demand in cancer cells (Haferkamp et al., 2020). Accordingly, we assume that cellular uptake of citrate increased during tumor growth.

Conclusions

Metabolic changes in cancer have long been acknowledged and considered as candidate therapeutic targets. The proposed mechanisms above remain hypothetical and need further investigation to validate, but the findings may provide for better diagnosis and prognostication for MM. We believe these results provide insight into mesothelioma metabolic reprogramming and may provide targets that can be exploited for therapeutic use.

Supplemental Information

Supplemental Information 1 ARRIVE Checklist

Click here for additional data file.

Supplemental Information 2 Ren Cell line

Click here for additional data file.

Supplemental Information 3 Cell Line Authentication

Click here for additional data file.

Supplemental Information 4 Differential SLC gene analyses. All ions detected by GC-MS and their relative peak intensities in all samples

FC: fold change; adj.P-Value: adjusted P-Value.

Click here for additional data file.

Supplemental Information 5 Survival associated genes in MM

Click here for additional data file.

Supplemental Information 6 Validation of expressions of SLC1A5 and SLC7A5 in MM

Immunohistochemical stains of SLC1A5 in MM CDX model (A) and MM patient (C); of SLC7A5 in MM CDX model (B) and MM patient (D).

Click here for additional data file.

Supplemental Information 7 GC-MS raw data

Click here for additional data file.

We thank International Science Editing for editing this manuscript.

Additional Information and Declarations

Competing Interests

Author Contributions

Animal Ethics

Data Availability

The authors declare there are no competing interests.

Yun Gao performed the experiments, analyzed the data, prepared figures and/or tables, authored or reviewed drafts of the paper, and approved the final draft.

Ziyi Dai analyzed the data, prepared figures and/or tables, authored or reviewed drafts of the paper, and approved the final draft.

Chenxi Yang analyzed the data, authored or reviewed drafts of the paper, and approved the final draft.

Ding Wang analyzed the data, prepared figures and/or tables, and approved the final draft.

Zhenying Guo performed the experiments, authored or reviewed drafts of the paper, and approved the final draft.

Weimin Mao conceived and designed the experiments, authored or reviewed drafts of the paper, and approved the final draft.

Zhongjian Chen conceived and designed the experiments, performed the experiments, analyzed the data, authored or reviewed drafts of the paper, and approved the final draft.

The following information was supplied relating to ethical approvals (i.e., approving body and any reference numbers):

Animal experiment in this research was performed under a project license [SYXK(Zhe)2017-0012, No. 2019-02-010] granted by the Institutional Animal Care and Ethics Committee of Zhejiang Cancer Hospital.

The following information was supplied regarding data availability:

The raw data are available in the Supplementary Table.

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
