# Peer review of "Metabolomics of a cell line-derived xenograft model reveals circulating metabolic signatures for malignant mesothelioma"

_PeerJ, doi:10.7717/peerj.12568_

## Round 0.1 · original submission · Major Revisions

The manuscript is original, however, issues mentioned by the Reviewers need to be addressed before it can be considered for publication.

·

Basic reporting

Gao et al propose a CDX model using metabolomics studies to identify drivers of mesothelioma. I find the concept very interesting and approach of the authors unbiased. I would have to propose some minor changes and additional experiments before the acceptance of this interesting work.

1. The language used through the paper is clear, but harsh in some places. The authors analyze the excising literature.

2. The structure of the figures is adequate.

Experimental design

I appreciate the fact that the authors reach these conclusions using an unbiased approach. Their question is well defined and the methods sufficiently detailed.

I have a couple of additional experiments/analyses that would support the proposed model.

Validity of the findings

1. The authors could also analyze the DepMap dependency map for their proposed genes https://depmap.org/portal/achilles/. The shown effect of the SLC genes in Mesothelioma cell Iines, will go a long way on supporting the proposed model.

2. The fact that some specific metabolites are differentially present in the proposed group, does not provide a direct link with their transporters. The authors should validate the expression of the proposed transporters in the CDX derived tumors.

3. It would be interesting to show the top differentially expressed genes in MM patients with high or low SLC signature. The analysis of one of the proposed genes is enough.

Additional comments

Gao et al propose a well supported model for markers of mesothelioma development. I would be happy to review a revised version of the manuscript. Good luck with the revision process.

Reviewer 2 ·

Basic reporting

no comment

Experimental design

see general comments

Validity of the findings

see general comments

Additional comments

The study of Gao and colleagues is focused on the identification of blood metabolites potentially associated with mesothelioma (MPM). They have applied mass spectrometry based metabolomics to investigate plasma samples from 10 xenograft mouse injected with tumor cells from the Ren MPM cell line and compared with 10 control mouse with no cancer. In general, the manuscript is well written and the methods are clearly described. Still, there are points which I would suggest to address for further clarification and cohesion of the message:
- Methodology:
o Looking into the literature, it seems that the original report about the MPM Ren cell line is from Smythe W.R. et al. (Smythe W.R., Hwang H.C., Amin K.M., Eck S.L., Davidson B.L., Wilson J.M., Kaiser L.R., Albelda S.M. Use of recombinant adenovirus to transfer the herpes simplex virus thymidine kinase (HSVtk) gene to thoracic neoplasms: an effective in vitro drug sensitization system. Cancer Res. 54:2055-2059, 1994)
o Line 196: what is the reason for reporting/mentioning about adenocarcinoma?
o Line 141, sample preparation: were the plasma samples pooled up to 50ul total volume or, rather, each sample was analyzed individually? How was normalization across the samples performed? What was the purpose of using an internal standard? Was this applied for normalization or for quantification or for others? What was the injected sample volume and were replicate injection used?
o How has been pathway enrichment analysis performed, resp. how is this done in the online platform https://www.metaboanalyst.ca/ ?
o Line 219, Figure 4a: what is the figure reporting? Is this a correlation based on circulating levels across all the samples? Relative circulation level of metabolite is relative to what? What is reported in the box-plots of figure 4?
o Line 180, bioinformatics analysis: what is the meaning of high and low rank samples (line 183)?
- Results:
o Line 226, dysregulated amino acid metabolism: what is the link of the amino acid results and the genes selected from the TCGA data? Why did the authors focus and select these genes?

- Discussion: the discussion section is verbous and could be shorter. More important, speculations are reported which are poorly supported by the results of the study or from literature evidence (e.g. line 253 hormons are unlikely the cause of cancer, line 261 “we suspect that …, line 286 “we suspect that the decrease in isoleucine…, etc. etc.). It would be helpful to report more evidence (from the data of the manuscript of from the literature) to provide evidence for the conclusions/interpretation

- Abstract: I would suggest to smooth the statement that MPM is not well studied because it does not reflect the major scientific efforts in the field

---

## Round 0.2 · Major Revisions

One of the reviewer feels that the concerns have not been adequately addressed and we provide you the opportunity to revise.

·

Basic reporting

Gao et al., return a significantly improved version of their already well controlled study.
I find the additional materials of high clarity and importance, overall improving the impact of the proposed work.
The English language has been significantly improved.
I would fully support the publication of this interesting work.

Experimental design

All the newly added experiments, are well controlled and adequately presented in the paper. No additional comments.

Validity of the findings

No comments, all the experiments are well controlled and supported by a combination of preclinical in vivo and clinical dataset models.

Additional comments

Congratulations to the authors for their effort.

Reviewer 2 ·

Basic reporting

The references should be reported at the right place in the manuscript

Experimental design

see additional comments

Validity of the findings

see additional comments

Additional comments

The authors have provided revisions on their manuscript. There still are major methodological as well as conceptual points to be addressed in order to support the data and the results, among them:
• Report in the method how normalization across samples was done.
• Report in the methods how quantitation of the metabolites was done and which strategy was used.
• Report in the methods if this was a pooled sample analysis (xenograft vs control) or if single animals were investigated.
• If single animals were investigated, report in the methods how data for each metabolites from each animal where handled and used for the analysis: average signal in xenograft-controls? Median? Ratio to internal controls? Ratio between groups? Etc. etc. .
• That following, report in the methods what was used for PCA etc..and what FC represents in the volcano plot and fold change of what.
• Explain in the manuscript how the data were retrieved from the XENA platform and which dataset was used and how these were combined with the dataset from the GEO database.
• Please report within the manuscript the definition of high/low rank samples.
• Figure 4 remains not clear. A: Is this a pearson correlation as reported in the method or a spearman as in the results? What do they correlate? Average or median values of the metabolite over the entire cohort (xenograft and controls)? Circulating levels are relative to what? Please explain in the text, within the method and in the results/figures. B-G: Please explain in the figure legend what the abbreviations are (e.g. FC), on what is the fold change based in the figure (average?median?others?), what are the whiskers and what are the bars. Please report in the methods how the P values are calculated.
• Please report the many prognosis-related genes from the bioinformatic analysis.
• What level of confidence has been used in table 3?
• Again, the link to why the authors have decided to focus on the SLC transporter genes is not provided. There is no experimental evidence within the manuscript to support the hypothesis that the levels of amino acid observed in the blood of the animal models may be related to the SLC transporter genes. The authors have decided to follow this hypothesis and explore a potential prognostic effect of their hypothesis, but they should guide the reader to what is the motivation of doing so and provide evidence for that. The “referenced paper” should be reported and referenced at the right place. They should also discuss this hypothesis in the discussion section. There is no evidence reported in the manuscript that these genes may be therapeutic targets.

---

## Round 0.3 · accepted · Accept

Thank you for addressing the reviewers' concerns.